# ComboPath: An ML system for predicting drug combination effects with superior model specification

**Duminda S. Ranasinghe**
Montai Health, Inc
Cambridge, MA, USA

**Nathan Sanders**
Montai Health, Inc
Cambridge, MA, USA
nsanders@montai.com

**Hok Hei Tam**
Montai Health, Inc
Cambridge, MA, USA

**Changchang Liu**
Montai Health, Inc
Cambridge, MA, USA

**Dan Spitz**
Montai Health, Inc
Cambridge, MA, USA

## Abstract

Drug combinations have been shown to be an effective strategy for cancer therapy, but identifying beneficial combinations through experiments is labor-intensive and expensive [Mokhtari et al., 2017]. Machine learning (ML) systems that can propose novel and effective drug combinations have the potential to dramatically improve the efficiency of combinatoric drug design. However, the biophysical parameters of drug combinations are degenerate, making it difficult to identify the ground truth of drug interactions even given experimental data of the highest quality available. Existing ML models are highly underspecified to meet this challenge, leaving them vulnerable to producing parameters that are not biophysically realistic and harming generalization. We have developed a new ML model, "ComboPath," aimed at a novel ML task: to predict interpretable cellular dose response surface of a two-drug combination based on each drugs' interactions with their known protein targets. ComboPath incorporates a biophysically-motivated intermediate parameterization with prior information used to improve model specification. This is the first ML model to nominate beneficial drug combinations while simultaneously reconstructing the dose response surface, providing insight on both the potential of a drug combination and its optimal dosing for therapeutic development. We show that our models were able to accurately reconstruct 2D dose response surfaces across held out combination samples from the largest available combinatoric screening dataset while substantially improving model specification for key biophysical parameters.

NeurIPS 2021 AI for Science Workshop.

# 1 Introduction

Drug combinations have emerged as a key technique for cancer therapy, yet discovering advantageous combinations through experimentation remains labor-intensive and costly. Biological testing of the effects of individual drug candidates is difficult and expensive and it is exponentially moreso for combinations. Computational approaches to increase the efficiency and effectiveness of combinatoric drug generation have so far been focused on, 1) interpreting complex and noisy experimental measurements for a given compound combination to infer the underlying bioactive properties of the formulation (i.e. "synergy scoring"; Ma and Motsinger-Reif [2019]) and 2) to nominate novel therapeutic combinations on the basis of known structure properties or biological effects of individual compounds (i.e. "combinatoric design"; Stathias et al. [2018]).

Unfortunately, progress in this field is stymied by the biological complexity of the problem. Synergy scoring and combinatoric design are both impacted by biophysical degeneracy expressed in experimental data, which exacerbates problems of model underspecification. Experimental data on two-drug combinations takes the form of 2D dose response data performed over a grid of doses; for example, each cell in the 2D array may represent the response of a cell (such as cell death, cytotoxicity) that has been exposed to a solution of each drug at a particular concentration. When the drugs are administered at high dose, the cellular response may increase because of the efficacy of the first drug, or the efficacy of the second drug, or a synergistic effect increasing the potency or cooperativity of the first drug induced by the presence of the second [Wooten et al., 2021]. These distinct biophysical effects can produce an equivalent, degenerate signal in the experimental dose response, and yet some are far more mechanistically plausible (consistent with biophysical expectations and commonly observed) than others and therefore likely to generalize well to new samples. Meanwhile, underspecification occurs when equivalent predictive performance can be achieved with a wide range of trained model configurations. This is a key challenge in ML that can lead to poor out of sample generalization [D'Amour et al., 2022]. Little prior work engages with the dose-dependent nature of combinatoric drug design, instead relying on point summaries of the dose response surface of drug combinations and exacerbating model underspecification.

In this work, we unify the fields of synergy scoring and combinatoric design with a single ML model, ComboPath, that has features addressing underspecification and data degeneracy. The key contributions of this work are 1) to propose the prediction of interpretable drug combination dose response surfaces as a novel ML task, 2) to introduce a new GNN-based model (ComboPath) capable of performing this task, 3) achieving greater model specification by integrating a biophysically-motivated model as an intermediate parameterization, and 4) leveraging domain knowledge to encode prior information that helps address degeneracy in observational data. This method represents the best in class tool for combinatorial drug design that can help investigators rapidly identify and screen combination therapy candidates. It provides the best approach to inferring the biophysical effects (synergy or antagonism) for denovo drug combinations and for optimizing dosing for a given cellular response or other experimental readout.

We review related work in §2 and present the architecture, design, and datasets used for our combinatoric drug design model in §3. In §4, we describe the results of our experiments with this model, showing the capability of our approach to simultaneously reconstruct dose response surfaces accurately while also aligning its latent parameterization of synergistic properties to domain knowledge. We conclude in §6.

# 2 Related Work

Our approach is inspired by the principles of AI that is explainable by design [Arrieta et al., 2020] and guided by theory [Karpatne et al., 2017]. In the field of combination drug therapy, the MuSyC (multi-dimensional synergy of combinations) model is a generalized, high dimensional Hill equation based on the laws of mass-action often used to describe the dose response surface of drug combinations [Wooten et al., 2021]. It uses 12 parameters to reconstruct the 2D dose response surface of a two-drug combination and each of the parameters represents an interpretable biophysical property such as the potency of the single agents or the synergy potential of the drug combination. Our model integrates the MuSyC model as a parametric description for the drug combination dose response surface, and by doing so, aligns the outputs of our high dimensional neural network to the interpretable biophysical properties of compound combinations to produce a cellular response in vitro.

Our work builds on top of recent advances in chemical representation learning. Owing to the graph-structured nature of small molecule chemical compounds and stemming from work by Hamilton et al. [2017], Mayr et al. [2018], Sun et al. [2019], Yang et al. [2019], and others, graph convolutional models such as the message passing neural network have achieved success in embedding chemical structure and predicting downstream tasks such as molecular properties and protein target binding affinities [Li et al., 2022]. Myriad other chemical embedding strategies exist for small molecules, such as transformers [Maziarka et al., 2020]. However, graph convolutional methods remain the most widely used and most competitive architectures in the field Ying et al. [2021].

Several previous works have proposed computational systems for combinatoric drug design and synergy prediction. The SynergySeq method of Stathias et al. [2018] calculates drug concordance scores by aggregating transcriptomic data, but is not supervised with combination data and is not capable of predicting dose response surfaces. Numerous supervised machine learning approaches have been developed to predict the synergy score of a drug combination on the basis of various characteristics of each drug [see for a detailed review Torkamannia et al., 2022]. Li et al. [2018] leveraged multi-modal experimental data to train shallow learning models to predict synergy scores. The Deepsynergy method of Preuer et al. [2018] applied a deep learning model to chemical descriptors for each drug to predict synergy scores. Jiang et al. [2020] formulates a similar task, predicting links in a graph of partially-known drug-drug interactions (where interaction edges reflect a binarized synergy score) using a graph convolutional network. Rather than predict an averaged synergy score, the model of Xia et al. [2018] predicts the specific cellular response at one particular dose combination; whichever concentration was measured to be the most efficacious for a given combination. Because these models are supervised with a one-dimensional summary of a dose response surface, they do not predict dose-specific responses and are vulnerable to model underspecification. Julkunen et al. [2020], Wang et al. [2021] have developed a system for predicting combination effects with flexible input feature specifications via latent factorization machines and latent tensor reconstruction. This system allows for prediction conditional on specific doses and supports imputation at doses not previously sampled via, e.g., polynomial regression. However, it does not supply interpretable biophysical parameters of the dose response surface and does not offer guarantees of regularization, particularly when predicting significantly out of the domain of training doses.

The ComboPath model extends previous work in several ways. It is the first model to address the novel ML task of interpretable dose response surface prediction. It is the first model to integrate a biophysical dose response surface function into a deep learning model, aligning the model specification to biophysically-interpretable parameterizations and regularizing to realistic response shapes. And it is the first model in this field to incorporate biophysical prior information to improve inductive biases, providing regularization on the dose response surface.

## 3 Methods

### 3.1 Problem formulation

We propose the novel ML task of 2D dose response surface prediction aligned to interpretable biophysical parameters. For a given pair of drugs $D_1$ and $D_2$ administered at doses $d1$ and $d2$, respectively, and a biological task $T$, we seek to predict the response surface, $E(d1, d2, T)$. $E$ represents any measurable biological response such as cytotoxicity or target-specific expression modulation in a given cell line and is predicted as a function of drug features, $f(D_i) \in \mathbb{R}^M$, and biological task features, $g(T) \in \mathbb{R}^N$. The drug features may be represented by an $M$-dimensional molecular embedding of chemical structure, embedding of known drug-target interactions, or any other featurization. The biological task may be any $N$-dimensional feature set encoding aspects of the experimental model, such as the genetic profile of a cell line, and one-hot encoding to differentiate between a variety of readouts, such as cytotoxicity or target-specific expression modulation. The ML task is formulated as a multi-task problem, where a single model is trained across multiple biological tasks.

### 3.2 Dose Response Datasets and Cross Validation

We trained our models on the drug combination cell viability screening data produced by O'Neil et al. [2016]. which screened combinations of 38 oncology drugs on 39 cancer cell lines on a $4x4$ dose matrix with 4 replicates per dose combination, producing a total dataset size of 22K dose response

surfaces and 1,475,332 individual cell viability measurements. We normalized each drug combination to the untreated condition to obtain fractional viability at each dose combination, which was used to fit the MuSyC parameters or directly train the model. In order to ensure symmetry during model training, we augmented the training set by flipping the order of the drug combination; this doubles the training set size and helps ensure equivalent predictions regardless of drug order. To test the performance of the model on novel drug combinations, we created a 10 fold cross validation set by leaving out drug pairs from the training data.

### 3.3 Compound and Cell Line Featurizations

Drug input to the model are featurized according to their known target interactions. Specifically, to represent a drug combination, for each drug the targets are mapped to a protein-protein interaction network and the perturbed pathways are inferred through a shared graph convolutional neural network. Cell lines are featurized using their basal RNAseq expression. An experimentally validated human protein-protein interaction network was extracted from the STRING database [Szklarczyk et al., 2022] as described by Gonzalez et al. [2021]. The protein targets of each compound were extracted from the STITCH database [Szklarczyk et al., 2016], CTDbase [Davis et al., 2021] and CHEMBL [Gaulton et al., 2017]. Each compound is represented by a graph which encodes the protein-protein interaction network. Each node on the graph denotes a protein on the protein-protein interaction network and each edge denotes an interaction between the two connecting proteins. The binary features of the node indicates whether the corresponding protein is targeted by the compound and as such the protein targets of the compound were mapped to the protein-protein interaction network.

We compiled and processed RNAseq basal expression data for 1000 cell lines from the Cancer Cell Line Encyclopedia data set [Ghandi et al., 2019]. We selected the 3,984 most diverse genes as the cell line featurization set. If the exact cell line used in a given dose response assay was not available in the DepMap dataset, the next closest cell line with according to cell type was selected.

### 3.4 Predictive Model Design

We develop two complementary implementations of our dose response surface prediction model "ComboPath" which incorporate a biophysically-motivated parameterization of the dose response surface to improves model specification. First, a "ComboPath-PS" model which is supervised on inferred synergistic properties of drug combinations and, second, a "ComboPath-RS" model that is supervised directly with individual dose response data points. These models are diagrammed in Figure 1.

#### 3.4.1 ComboPath-PS: Parameter Supervised

A set of 12 MuSyC parameters were individually fitted to each dose response surface via the procedure of S3.5 and used as labels for training. Outputs from the compound graph convolution layers and cell line features are passed separately into two fully connected neural networks to obtain the inferred combination and cell line representations. These representations are then passed through a 3-layer fully connected neural network to predict the 12 MuSyC parameters.

#### 3.4.2 ComboPath-RS: Response Supervised

The compound features and cell line features were processed through two separate fully connected layer sets as described in §3.4.1. Each of the inferred compound representations is combined with the cell line representation and passed through a fully connected neural network ("single compound processor") to output the parameters describing single compound activity ($C$, $h$, and $E$). The representations of two compounds are then combined along with the cell line representation to predict interactive parameters (e.g. $\gamma_{12}$) via a fully connected neural network ("combo processor"). These 12 parameters comprise the MuSyC representation and form the last hidden layer of the model and, instead of becoming the primary output of the model or used directly for supervision, these dose response surface model parameters are used to calculate the dose-specific effects of a combination for a given cell line. As a result, the 2D dose response function effectively organizes the last hidden layer in a biophysically meaningful way–each number in the last hidden layer is a parameter in the MuSyC function that denotes properties of the combination such as the individual drug EC50 ($C1/C2$), combination cooperativity, and maximum effect ($E3$).

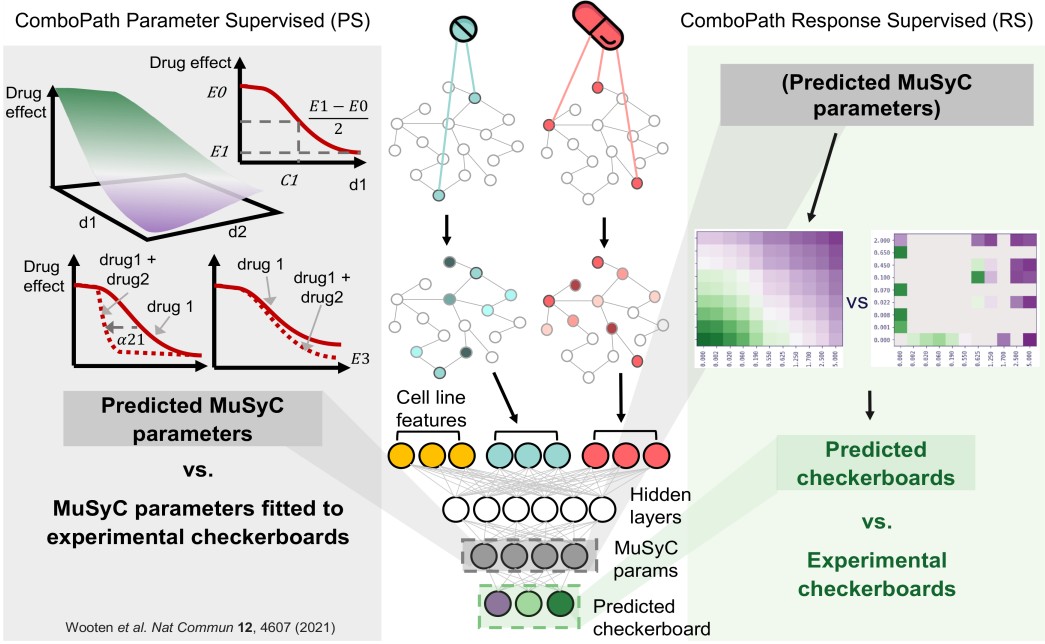

Figure 1: Comparison of the ComboPath-PS and ComboPath-RS models. Both models take the same combination of compound and cell line features as input. While the ComboPath-PS model outputs the 12-dimensional parameterization of the 2D dose response surface, the ComboPath-RS model directly outputs the predicted effect at given doses of the compounds.

Both the ComboPath-RS and ComboPath-PS models are explainable by design in that they generate dose response surfaces via a directly interpretable parameterization, although this work does not directly address the interpretation of the parameters of the graph representation learning layers of the models.

### 3.4.3 Incorporation of Prior Information in ComboPath-RS

During model training, we imposed prior information on the intermediate biophysical parameterization of the model by adding a NLL Gaussian penalty term to the loss integrating a Gaussian prior distribution for each MuSyC parameter. We develop a set of Half-Gaussian priors for the MuSyC parameters $\Psi$ that reflect biophysical domain knowledge about combination dose responses surfaces, specified in Table 1. We fix the $E_0$ parameter at identically 1, reflecting the normalization that untreated cells have no drug-induced cytotoxicity. We use a multivariate prior on the $C1/C2$ and $E1/E2$ parameters, for which the 2D dose response function induces strong posterior covariance:

For the purpose of ablation testing, we also fit a version of ComboPath-RS without prior information. These prior distributions were built and iterated on over time based on observations about the goodness of posterior fits. In particular, we determined that informative priors were helpful to regularize parameters that are not well-identified by the experimental data, such as slope parameters that are effectively unobserved when a compound is ineffective within a given cell line.

### 3.5 Establishing ground truth biophysical parameters

To establish ground truth[1] dose response surface parameters for compound combinations, we employ a hierarchical Bayesian modeling methodology fit using a Hamiltonian Monte Carlo sampling

---

[1]Herein we use "ground truth" to refer to the biophysical dose response surface parameters that are inferred by fitting the MuSyC model directly to experimental biological data, as opposed to those parameters predicted for a de novo compound combination by the ComboPath model.

technique [Hoffman et al., 2014]. We implement the 2D dose response surface function of the MuSyC model in the Stan modeling language [Carpenter et al., 2017]. We apply a Gaussian likelihood function for the model that compares this parameterized dose response function together with a heteroskedastic variance parameter that is a function of the dose grid, $E_{c1,c2,l,r}(d1, d2) \sim$ N $(\text{MuSyC}(\Psi, d1, d2), \sigma(d1, d2))$, where $E$ is the experimental measurement of a single biological activity measurement (nominally ranging from $0 - 100\%$) corresponding to one combination of compounds $c1$ and $c2$, for one cell line $l$, for one experimental readout $r$, at one dose pair $d1$ and $d2$ defined in molar units; $N$ represents the Gaussian distribution; MuSyC is the MuSyC dose response surface model; $\Psi$ is the vector of 12 biophysical parameters that specifies the dose response surface; and $\sigma$ is the variance parameter. In particular, we choose a step-function form of $\sigma$ that reduces the variance allowable for single-compound samples (where either $d1$ or $d2 = 0$) by a factor of 9, reflecting the lower error that typically accompanies the less-experimentally-complex single compound measurements. The prior distributions are specified in Table 1.

## 4 Results

### 4.1 Reconstruction of 2D response surface

We first tested if the ComboPath models can reproduce the experimental results at specific dose combinations, i.e. to reconstruct the dose response surface of cytotoxic effects of held out drug combinations reported by O'Neil et al. [2016]. To visualize the behavior of the model, we plotted the experimental and predicted dose response surfaces from a random drug combination (lapatinib, x-axis, and dasatinib, y-axis, on the cell line SKMES1). The experimental dose response surface is sparse in the original dataset (Figure 2, "Experimental"). For comparison, we show the ground truth 2D dose response surface interpolated from MuSyC parameters directly fitted to experimental dose response surfaces (Figure 2, "Theoretical limit"). For the ComboPath-PS model, we used the predicted MuSyC parameters to calculate the viability of the cells at a series of doses (Figure 2, "ComboPath-PS") and, for the ComboPath-RS model, we directly plotted the prediction from the model at the same given doses with and without prior information applied during training (Figure 2, "ComboPath-RS no prior" and "ComboPath-RS with prior"). In this example, the dose response surface reconstructed from the ComboPath-PS model is an excellent match to the experimental data and the ground truth direct fit, as is the ComboPath-RS model with priors applied. The ComboPath-RS model without priors under-predicts the efficacy of the first drug (lapatinib), which leads it to also underpredict their joint effect at maximal combined dose.

We evaluate the overall predictive performance of the ComboPath models by calculating the mean absolute error (MAE) of the predicted bioactivity level across dose combinations for each drug combination in Figure 3 (summary statistics are reported in Table 2). To reflect the theoretical upper bound on predicted model performance, we display the results for models directly fitted to the experimental data. Assuming the biophysically-motivated MuSyC model truly represents the theoretical range of cellular responses to combinatoric therapies, these direct fits represent the highest possible performance of our predictive models on held out combinations, with non-zero MAE associated with only biological variability and experimental error. All three ComboPath models have median MAE 13%, approaching the theoretical limit of 8%. We find that the ComboPath-PS model achieves the lowest median MAE, but has the strongest tail and therefore the highest MAE standard deviation. In other words, this model has the best predictive performance in the typical case, but more often suffers from major predictive errors on some combinations than the other models. The ComboPath-RS model without prior information performs similarly well in the typical case (similar median), but has superior mean MAE of 14% versus 18% for ComboPath-PS (fewer major errors).The ComboPath-RS model with prior information has very similar MAE distributional characteristics to the ComboPath-RS model without prior information; in the next section we will consider its parameter inference performance.

### 4.2 Improvement of model specification, regularization, and degeneracy

We compared the ground truth MuSyC parameters against the model-predicted parameters (Figure 4). As expected because the ComboPath-PS model used the experimentally-fitted parameters as training labels, the ComboPath-PS model showed the highest correlation between grund truth and predicted parameters. The ComboPath-RS no prior model inferred values that poorly matched the experimental

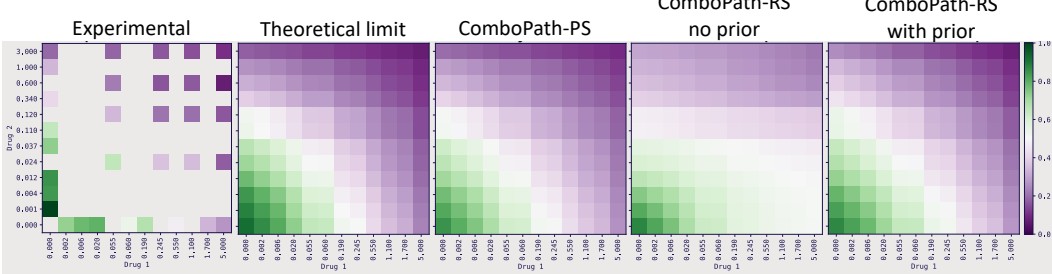

Figure 2: Example reconstructed dose response surfaces. Color bar indicates fraction of viability compared to untreated cells. Experimental: viability measurements at sparsely sampled doses reported by O'Neil et al. [2016]; Theoretical limit: surfaces interpolated by directly fitting "ground truth" MuSyC parameters to the experimental surfaces; ComboPath-PS: response surface reconstructed from ComboPath-PS-predicted MuSyC parameters on the held out drug combination; ComboPath-RS no prior: the predicted effects at densely sampled doses for the held out combination produced by the ComboPath-RS model without prior information used in model training; ComboPath-RS with priors: the predicted effects for the held out combination predicted by the ComboPath-RS model with prior information imposed on the MuSyC parameters.

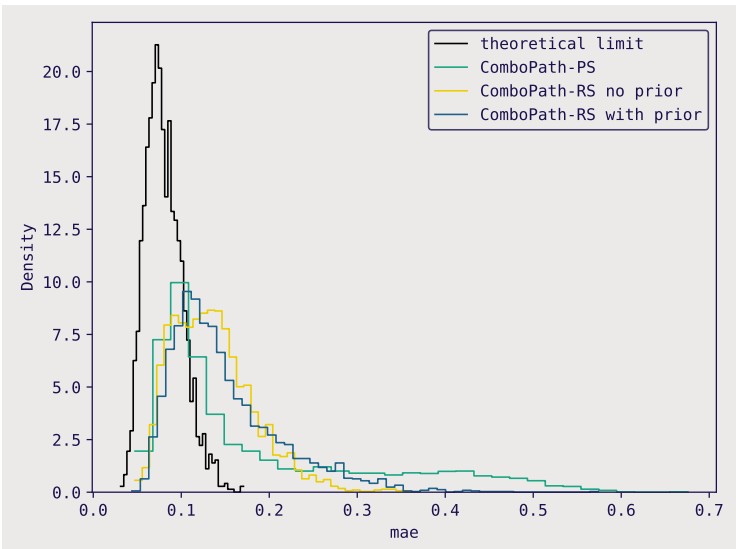

Figure 3: Overall performance of ComboPath models. The Mean Absolute Error (MAE) between the predicted and experimental dose response surface was calculated for each pair. The density plots showed the distribution of the MAE in the test set. Y axis shows the MAE of fraction viability compared to untreated cells. Black distribution indicates the theoretical maximal performance as it is the distribution of the MAE by directly fitting a set of parameter to the experimental surface.

parameters; it sometimes yields extreme intermediate parameters that may reproduce observed dose response samples, but fail to generalize.

When prior information was imposed, the correlation on $\log C1$ (which reflects the potency of an individual compound within the combination) increased from 0.36 in the ComboPath-RS no prior model to 0.78 in the ComboPath-RS with prior model. Figure 4 shows that the ComboPath-RS with prior model improved the correlation by fixing a bias in the ComboPath-RS no prior model which consistently underestimates the values of this parameter (and, equivalently for $\log C2$. We note also that the performance on $\log C$ and other individual compound parameters (maximal efficacy,

$E1$ & $E2$, and individual cooperativity, $h1$ & $h2$) is high across all models, due to memorization of individual compound effects as the models have seen some compounds from the test set during training in combination with other compounds.

The models were more challenged in predicting interactive parameters. The ComboPath-RS models have lower performance than ComboPath-PS in predicting $E3$ (the maximal effect of the combination) and $\gamma_{12}$ (cooperativity of the combination), yielding correlations $\lesssim 0.3$ versus $\sim 0.7$. As the models still achieved good performance in reconstructing the dose response surfaces, this reflects degeneracy in the interactive parameters: disparate parameter values that can generate similar dose-specific effects. For some parameters, ComboPath-RS with priors effectively circumvents this degeneracy by marginalizing over the intermediate parameterization with prior information about parameter values. In particular, ComboPath-RS with priors achieves better reconstruction of the $\gamma_{12}$ parameter (correlation $\sim 0.3$), whereas the ComboPath-RS model without prior results in zero correlation and infers extreme values ($\gamma_{12} \sim 5$) that do not exist in the ground truth results.

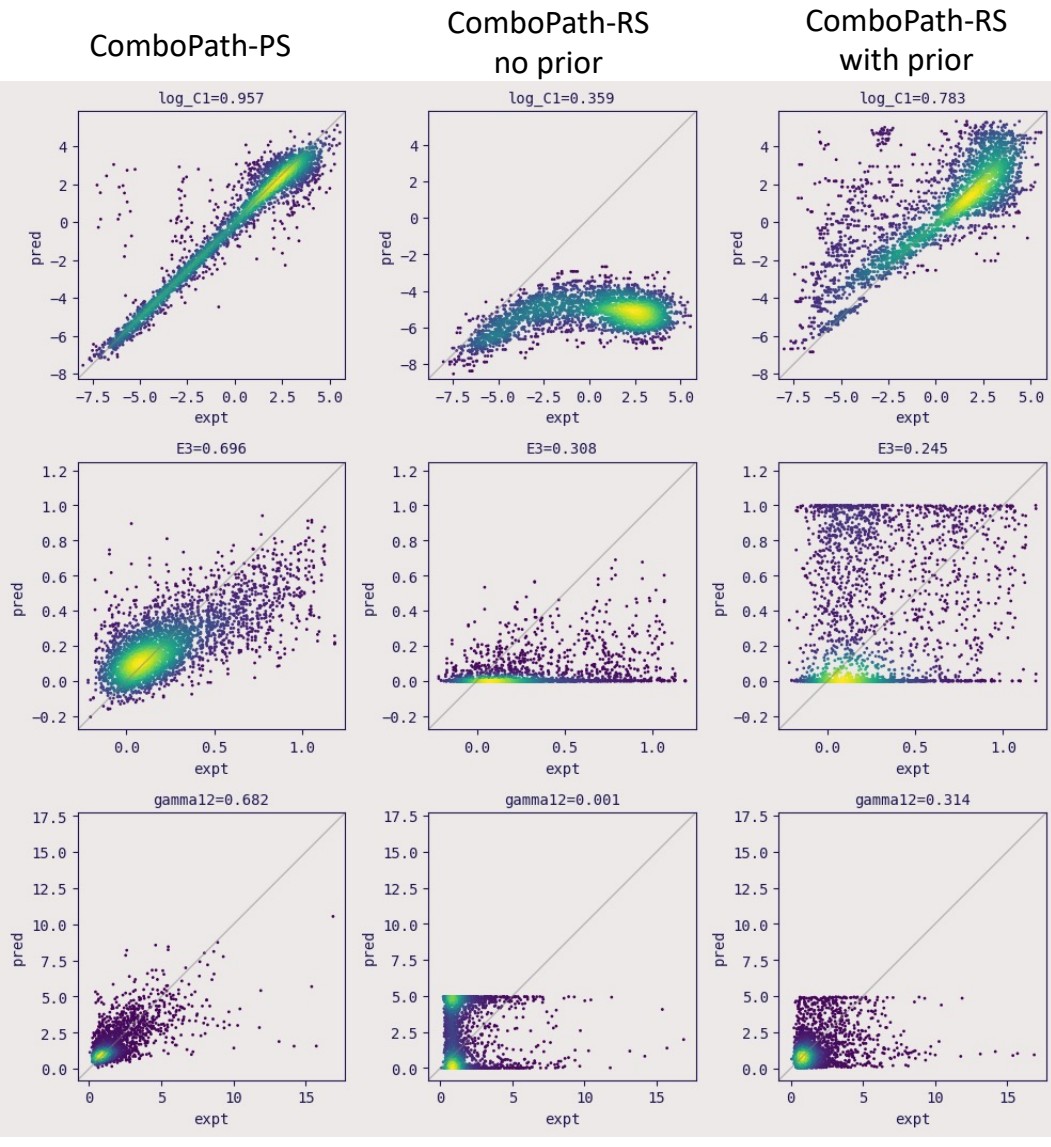

Figure 4: Scatter plots of ComboPath-predicted MuSyC parameters for held out combinations (y-axis) vs. ground truth MuSyC parameters directly fitted to experimental dose response surfaces (x-axis). The facet titles report the Pearson correlation.

# 5    Discussion

We developed a new ML model, ComboPath, to solve the novel task of predicting interpretable 2D dose response surfaces for drug combinations that has superior model specification. Investigators can prioritize and nominate drug combinations for experimental validation based on ComboPath-predicted effects at any optimal dose, or based on any summary metric (such as ZIP, Lowe, CI, or MuSyC $\beta$). Although this work presents results on large scale public oncology datasets, the model can be applied in other disease areas such as immunological, infectious, and metabolic diseases where the combination data is more limited, because of its data efficiency and improved model specification, and could also be generalized to 1D or $K > 2$ dimensional dose response spaces.

The model is data efficient as it can incorporate data from across experiments while learning from every datapoint. Drug response, compound-protein interaction, and combinatoric effect prediction require diverse data to supervise training across several dimensions: chemical space, chemical interactions (combinations), dose, and biological tasks (e.g. cell lines and cellular responses). This high dimensionality makes dense experimental sampling impossible. ComboPath's versatility is a new strategy for combinatoric modeling, integrating disparate datasets with different experimental setups and training simultaneously across multiple dimensions, thereby improving generalization. For a given biological task, ComboPath can train simultaneously on 1) small combinatoric datasets densely (or sparsely) sampling dose response surfaces on lead molecules and 2) public high throughput screening (HTS) datasets generated at fixed dose on the same assay. And ComboPath can train across multiple biological tasks simultaneously, providing better sampling of biological task space. Whereas previous synergy prediction approaches collapse information across all samples onto one data point (e.g. an average ZIP score), ComboPath leverages data from every dose sample and replicate.

ComboPath has regularizing effects suitable for small datasets. The adoption of the parametric form of the dose response surface improves model specification and prevents the prediction of non-physical dose response surface shapes. For example, dose "hotspots" that may appear due to aleatoric noise in experimental data that make one dose combination appear significantly more effective than neighboring doses are smoothed out in our fitted models. While there is some risk of model mis-specification that could enhance epistemic uncertainty when using the parametric intermediate [Kato et al., 2022], a response shape resembling a 2D generalization of the Hill curve is widely observed in combinatoric drug screening [see e.g. Lehár et al., 2007, Wooten et al., 2021, Ianevski et al., 2022]. The model is additionally trained with experimental replicate-level data, supervising with the measurement label distribution, which provides implicit regularization [Song et al., 2022].

While the model presented here leverages a 2D dose response curve to predict combination effects, this parametric approach can be easily generalized to the activation/inhibition effect on a protein target or other dose response of $K = 1$ (single compound) or $K > 2$ (multiple compound) combinations.

# 6    Conclusion

We have introduced ComboPath, a novel GNN-based ML model for designing drug combination therapies and demonstrated its potential using a combinatoric screen of cancer therapeutics. We have shown that our approach to improving model specification by incorporating biophysically-meaningful parameterizations and prior information is effective at aligning synergistic properties inferred by the model to biophysical domain knowledge, while achieving high predictive accuracy at the novel task of  reconstructing dose response surfaces with interpretable parameters. ComboPath represents a new ML approach to accelerate drug design, increasing the efficiency of combinatoric search in therapeutic discovery.

## Acknowledgments and Disclosure of Funding

We thank Connor Coley, many colleagues at Montai Health, and the anonymous reviewers for valuable feedback that informed this work.

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

# A  Prior distributions over MuSyC parameters

We developed a set of Half-Gaussian priors for the MuSyC parameters $\Psi$ that reflect biophysical domain knowledge about combination dose responses surfaces, specified in Table 1. These are used directly to regularize the ComboPath-RS model as well as in establishing the ground truth parameter values used to supervise the ComboPath-PS model, as discussed in more detail in §3.5 and §3.4.3.

Table 1: Half-Gaussian prior distributions adopted for MuSyC parameters

| MuSyC parameter | Prior mean | Prior std. dev. | Annotation |
|---|---|---|---|
| $\sigma_0$ | 0 | 0.1 | Response variance parameter |
| $E3$ | 0 | 1 | Maximal efficacy of the combination drugs 1 and 2 |
| $h1, h2$ | 0 | 5 | Hill coefficients for 1D dose-reponse curve of drug 1 and 2 isolation |
| $\log(\alpha_{12})$, $\log(\alpha_{21})$ | 0 | 1 | Log fold change in the potency drug 2 by drug 1, log fold change in the potency drug 1 by drug 2 |
| $\log(\gamma_{12})$, $\log(\gamma_{21})$ | 0 | 1 | Log fold change in the potency drug 2 by drug 1, log fold change in the potency drug 1 by drug 2 |
| $(\log(C1),\ E1)$, $(\log(C2),\ E2)$ | $\begin{bmatrix} \log(100) \\ 0 \end{bmatrix}$ | $\begin{bmatrix} \log(100) & -0.75 \\ -0.75 & 1 \end{bmatrix}$ | Multivariate prior over: $E$, Maximum efficacy of drug 1 and 2 in isolation and $C$, Concentration of drug need to achieve 50% of maximum effect (AC50). |

# B  Model performance summary

We report the performance statistics from our tested models in tabular form in Table 2. All reported values are statistics of the MAE (mean absolute error) distribution shown in Figure 3 for the predicted cytotoxicity response across compound combinations, cell lines (lower is better). Results are truncated to the typical precision of the SEM (standard error of the mean).

Table 2: Performance summary for ComboPath models.

| Model | mean | median | std | SEM |
|---|---|---|---|---|
| ComboPath-PS | 0.1841 | 0.1252 | 0.1268 | 0.0022 |
| ComboPath-RS no prior | 0.1360 | 0.1306 | 0.0468 | 0.0008 |
| ComboPath-RS with prior | 0.1499 | 0.1333 | 0.0632 | 0.0011 |
| Theoretical limit | 0.0803 | 0.0775 | 0.0212 | 0.0003 |

