# OpenReview forum: "ComboPath: A model for predicting drug combination effects"
_NeurIPS.cc/2023/Workshop/AI4Science — NeurIPS2023-AI4Science Poster_

### Official Review · Reviewer_SEtJ · 2023-10-24
**Review of ComoPath**

**Rating:** 5
**Confidence:** 2

**Review:**

"biophysical parameters of drug combinations are degenerate"
- you may want to explain this more

1) to propose the prediction of drug combination dose response surfaces as a novel ML task,
2) to introduce a new GNN-based model (ComboPath) capable of performing this task,
3) achieving greater model specification by integrating a biophysically-motivated model as an intermediate parameterization, and
4) leveraging domain knowledge to encode prior information that helps address degeneracy in observational data. This method represents the best in class tool for combinatorial drug design that can help investigators rapidly identify and screen combination therapy candidates


The authors claim this is a novel task. However, this seems like it has been done in ComboFM? Can the authors address this?
https://www.nature.com/articles/s41467-020-19950-z

"If the exact cell line used in a given dose response assay was not available in the DepMap dataset, the next closest cell line according to cell type was selected."
-- can the authors quantify how close these replacements are?

It would be useful if the authors can further clarify why they chose their ground truth methodology. Shouldn't direct experimental values be used, instead of a sampling process? They fit psi values to experimentally determined sparse grids. However, this requires the MuSyC model to be valid in reality. If nothing else, since the authors claim to be proposing a new task, this part should be explained in more detail in section 3.5. In particular, I wonder why not use actual dose-response values from the NCI-ALMANAC, like is done in ComboFM? Wouldn't it be better to compare against direct experimental even if it is a sparse matrix?
-- It seems the theoretical upper bound is from fitting models directly to experimental data. The authors use the MuSysC model as a ground truth, but then they claim that MuSysC models fit to experimental data are a theoretical upper bound. Doesn't this make using it as a ground truth a flawed measurement with MAE?
-- In Figure 3, the authors show MAE density between the predicted and experimental checkerboard. I believe this is using the experimental data, which is what I asked for above, but then it is unclear what "ground truth" means previously.



"We display the mean absolute error distribution of the to To reflect the theoretical upper bound on predicted model performance, we display the results for models directly fitted to the experimental data"
--unclear and ungrammatical

"All three ComboPath models have MAE modes 10%, only a few percent larger than the theoretical limit."
-- unclear

The authors measure performance using MAE. However, they also discuss differences in the long-tail performance on rare cases. Is there a better metric that can be used to report this? Right now, analysis seems very qualitative. e.g., "with a mode very similar to the ComboPath234 PS model, but a superior tail".


"§??)" -- should be fixed

I'm not sure "(C, h, and E)" on line 161 are defined anywhere.

I think it would be interesting to talk more about "degeneracy" discovered in the proposed models. This work claims to address degeneracy in the introduction-- how does this finding reconcile with that?

" public high throughout screening"
-- typo

"The model is additionally trained with experimental replicate-level data,"
-- can this be clarified?

Following the directions of the area chair, I did not review page 9.

The model architecture appears to be a MLP where the final hidden layer is supposed to represent the 12 parameters of MuSysC.

The only discussion of the loss I could find was "adding a NLL Gaussian penalty term to the loss ".

Caption 2 refers to a "Direct fit:" which is not in the actual figure.


Overall, I think the proposal to fit a model based on MuSysC is quite interesting. It helps improve explainability of the deep model, although it only does so by predicting biologically important parameters for MuSysC, so this is not a full explanation.

I found comparisons between methods to be poorly done. Figure 3 appears be the closest useful comparison between the 3 proposed models. However, it is still just plotting different error densities in the same figure; I believe more quantification is needed. I believe it would be better to have compared against additional baselines, even if they are not based on the MuSysC paramterized surface. Can different standard error metrics be reported in a table? The idea of a "ground truth" is also unclear.

Standard ML baselines would have been useful to see, such as random forest. Further, the authors claim this is a new task. They discuss Xia et al. 2018 like I expected, but they don't discuss ComboFm's tensor factorization approach (Julkunen et al. 2020). It is unclear to me why ComboFm cannot serve as a baseline here. The paper also has several cases where the writing is unclear, as detailed above. It also does not follow the CFP by going onto a 9th page. I would recommend the authors to work on the readability of the paper and to take advantage of appendix space for further details when necessary. Different terms are used throughout the paper to refer  to the same thing. I would also recommend them to perform more standard quantitative analysis as is typically done in ML papers. The training procedure is also not well specified. What loss function was used? How many epochs was training. How many parameters was the model?

Despite these concerns, I believe the underlying idea is solid. I believe more extensive evaluation, improved writing, and further details would greatly benefit the presentation of this work to an outside audience. Further, I would encourage the authors to be more clear/extensive in their discussion about why this is a new task and why they don't try other models as baselines, at least for predicting psi values. There is the possibility I am missing ideas from related work which would improve clarity.